# Ni-catalyzed hydroaminoalkylation of alkynes with amines

Wei-Wei Yao[1], Ran Li[1], Hao Chen[1], Ming-Kai Chen[1], Yu-Xin Luan[1✉], Yi Wang [2], Zhi-Xiang Yu [2✉] & Mengchun Ye [1✉]

Allylic amines are versatile building blocks in organic synthesis and exist in bioactive compounds, but their synthesis via hydroaminoalkylation of alkynes with amines has been a formidable challenge. Here, we report a late transition metal Ni-catalyzed hydroaminoalkylation of alkynes with N-sulfonyl amines, providing a series of allylic amines in up to 94% yield. Double ligands of N-heterocyclic carbene (IPr) and tricyclohexylphosphine (PCy₃) effectively promote the reaction.

[1] State Key Laboratory and Institute of Elemento-Organic Chemistry, College of Chemistry, Nankai University, Tianjin, China. [2] Beijing National Laboratory for Molecular Sciences (BNLMS), Key Laboratory of Bioorganic Chemistry and Molecular Engineering of Ministry of Education, College of Chemistry, Peking University, Beijing, China. ✉email: yxluan@nankai.edu.cn; yuzx@pku.edu.cn; mcye@nankai.edu.cn

Allylic amines not only widely exist in a broad range of natural products and bioactive compounds, but also serve as versatile building blocks in organic synthesis[1–4]. The development of efficient and general methods for their synthesis has received much attention during the past several decades[5–23]. Among various reported methods, transition metal-catalyzed hydroaminoalkylation of π-unsaturated compounds, such as alkenes and alkynes represents one of the most straightforward and atom-economical synthetic routes[24–31]. With using either early[24–26] or late transition metals as catalysts[27–31], a large number of hydroaminoalkylations of alkenes have been developed. However, in sharp contrast, analogous reactions of alkynes were faced with tremendous challenges (Fig. 1a), likely owing to difficult alkyne insertion and challenging protonolysis[32–39]. A pioneering investigation was conducted in 1989 by Buchwald and co-workers, who successfully obtained the allylic amine product by aqueous work-up. Despite an efficient protonolysis, the regeneration of Zr catalyst cannot be realized in this protocol, leading to stoichiometric Zr-complex needed[32]. Since then, much effort has been devoted to improving the reaction[33–39], while the development of a catalytic method has been an elusive challenge. Most recently, during our submission, Schafer group used a tetradentate bis(ureate) ligand and metal Zr to in situ form a bulky Zr catalyst, achieving a catalytic hydroaminoalkylation of alkynes for the first time (Fig. 1b)[40,41]. The bulky ligand proved critical to the reactivity, not only facilitating alkyne insertion, but also allowing the coordination of neutral amines to the metal center for subsequent easier protonolysis. Despite this big advance, the early transition metal Zr-catalyzed method still suffered from some undesired limitations such as unavoidable hydroamination side reaction in many cases, difficult-to-remove N-aryl protecting groups, and tricky regioselectivity under relatively harsh conditions. Therefore, the development of other efficient catalytic systems for hydroaminoalkylation of alkynes is still highly desirable. In this work, we use an inexpensive nickel as a catalyst to achieve a late transition metal-catalyzed hydroaminoalkylation of alkynes with N-sulfonyl amines, providing a series of allylic amines in up to 94% yield (Fig. 1c). The reaction features relatively mild conditions (80 °C), general substrate scope of both amines and alkynes and high regioselectivity.

## Results

**Reaction optimization.** In comparison with various hydroaminoalkylations of alkenes, the difficulty of hydroaminoalkylation of alkynes was ascribed to the following possible reasons: (1) strong basic and nucleophilic alkyl or aryl amines could coordinate to metal centers, resulting in either deactivation of transition metals or undesired side reactions such as hydroaminations; (2) weak acidity of N−H bonds cannot effectively undergo protonolysis of metallocycle intermediates. Thereby, the selection of proper N-protecting groups to increase the acidity of amines could be critical to the reaction efficiency, because more acidic amine would significantly reduce its coordination with metal centers and other side reactions such as hydroamination. However, to accommodate acidic N−H bonds, sensitive early transition metal complexes should be replaced by late transition metals such as Pd, Ru and Ni, because they could have better compatibility with protic substrates and solvents.

Following this hypothesis, we conducted an extensive survey on N-protecting groups, transition metals, ligands and other reaction parameters. Ultimately, triisopropylbenzenesulfonyl

**Fig. 1 Transition metal-catalyzed hydroaminoalkylation of alkynes. a** Pioneering investigation using stoichiometric Zr-complex (Buchwald). **b** Strategy I using early transition metal Zr and bulky tetradentate ligand (Schafer). **c** Strategy II using late transition metal Ni with dual ligands (this work). IPr = 1,3-bis (2,6-diisopropylphenyl)-2,3-dihydro-1H-imidazole. Cy₃P = tricyclohexylphosphine.

**Fig. 2 Reaction optimization.** Reaction conditions: **1a** (0.20 mmol), **2a** (0.22 mmol), toluene (2.0 mL) under $N_2$ for 1 h; yield was determined by $^1$H NMR using $Cl_2CHCHCl_2$ as the internal standard. $IPr^{Me}$ = 1,3-bis(2,6-diisopropylphenyl)-4,5-dimethyl-2,3-dihydro-1$H$-imidazole. SIPr = 1,3-bis(2,6-diisopropylphenyl)imidazolidine. IMes = 1,3-dimesityl-2,3-dihydro-1$H$-imidazole. dppe = 1,2-bis(diphenylphosphino)ethane.

| entry | deviation from the standard conditions | | yield of **3a** (%) |
|---|---|---|---|
| 1 | *no deviation* | | **99** |
| 2 | TPS replaced by | 2,4,6-Me$_3$-C$_6$H$_2$SO$_2$ (TMS) | 9 |
| 3 | | *p*-Me-C$_6$H$_4$SO$_2$ (Ts) | trace |
| 4 | | *p*-CF$_3$-C$_6$H$_4$SO$_2$ | 7 |
| 5 | | *p*-MeO-C$_6$H$_4$SO$_2$ | trace |
| 6 | | CH$_3$-SO$_2$ (Ms) | 7 |
| 7 | | CF$_3$-SO$_2$ (Tf) | 0 |
| 8 | | $^t$Bu-SO$_2$ (Bs) | 0 |
| 9 | IPr·HCl replaced by | 0 | 0 |
| 10 | | IPr$^{Me}$·HCl | 76 |
| 11 | | SIPr·HCl | 20 |
| 12 | | IMes·HCl | 34 |
| 13 | PCy$_3$ replaced by | 0 | 14 |
| 14 | | Ph$_3$P | 51 |
| 15 | | $^t$Bu$_3$P | 17 |
| 16 | | $^n$Bu$_3$P | 11 |
| 17 | | dppe | 17 |
| 18 | Ni(cod)$_2$ replaced by | 0 | 0 |
| 19 | | NiCl$_2$·diglyme | 0 |
| 20 | | NiCl$_2$·diglyme with Mn | 0 |

(TPS) was identified as the superior N-protecting group and Ni/IPr/PCy$_3$ was identified as the optimal catalyst. With their combination, hydroaminoalkylation of alkyne **2a** with N-TPS amine **1a** smoothly proceeded under mild conditions (80 °C), providing the corresponding allylic amine **3a** in nearly quantitative yield (Fig. 2, entry 1).

Control experiments showed that the alteration of TPS resulted into significantly diminished yields (entries 2−8). For example, the replacement of isopropyl groups (TPS) by methyl groups (TMS) gave only 9% yield (entry 2). Common *p*-tolylsulfonyl (Ts) further decreased the yield to a trace amount (entry 3). The combination of NHC (IPr) and phosphine (PCy$_3$) ligands also proved critical to the reaction (entries 9−17). The absence of IPr·HCl completely inhibited the reaction (entry 9), whereas the reaction still gave **3a** in 14% yield without the addition of PCy$_3$ (entry 13), demonstrating the vital role of IPr and the promoting effect of PCy$_3$. In fact, a yield of 68% was detected with IPr alone at an elevated temperature (110 °C) but with poor reproducibility (see the Supplementary Information for details). Other carbenes and phosphines were less effective (entries 10−12 and 14−17). Without Ni(cod)$_2$ or with other nickel species, the reaction did not work (entries 18−20).

**Scope of amines and alkynes.** Under the optimized conditions, various N-TPS amines were then examined (Fig. 3). Results showed that the reaction tolerated a broad range of functional groups on the phenyl ring of N-benzylamines, including simple alkyl (Me, **3b**−**3d**), electron-donating groups (alkoxy, **3e** and **3f**), and electron-withdrawing groups (OCF$_3$, F, Cl, CF$_3$, CN, and CO$_2$Me, **3g**−**3n**), providing the corresponding allylic amines in 62−94% yields. In addition, the position of substituents did not have a strong influence on the reaction yield (**3b**−**3d** and **3h**−**3j**). Notably, both 1-naphthyl (**3o**) and heteroaryl (**3p**) instead of the phenyl of **1a** also worked well, affording both 86% yields. When the phenyl was replaced by the alkenyl, a decreased yield was obtained (45%, **3q**) in the presence of 10 mol% of the catalyst at 110 °C. Notably, various N-alkylamines were still compatible with the reaction (**3r**−**3u**, 41−54% yields), but requiring harsher conditions (130 °C and 20 mol% catalyst) and a Ts protecting group. We reasoned that higher α-C–H bond strength of alkylamines than that of benzylamines and higher activation energy may result in this situation.

Next, a broad range of alkynes were investigated under the standard conditions (Fig. 4). Various diaryl alkynes bearing alkyls (**4a**−**4e**) and electron-donating groups (**4f**) on the phenyl rings

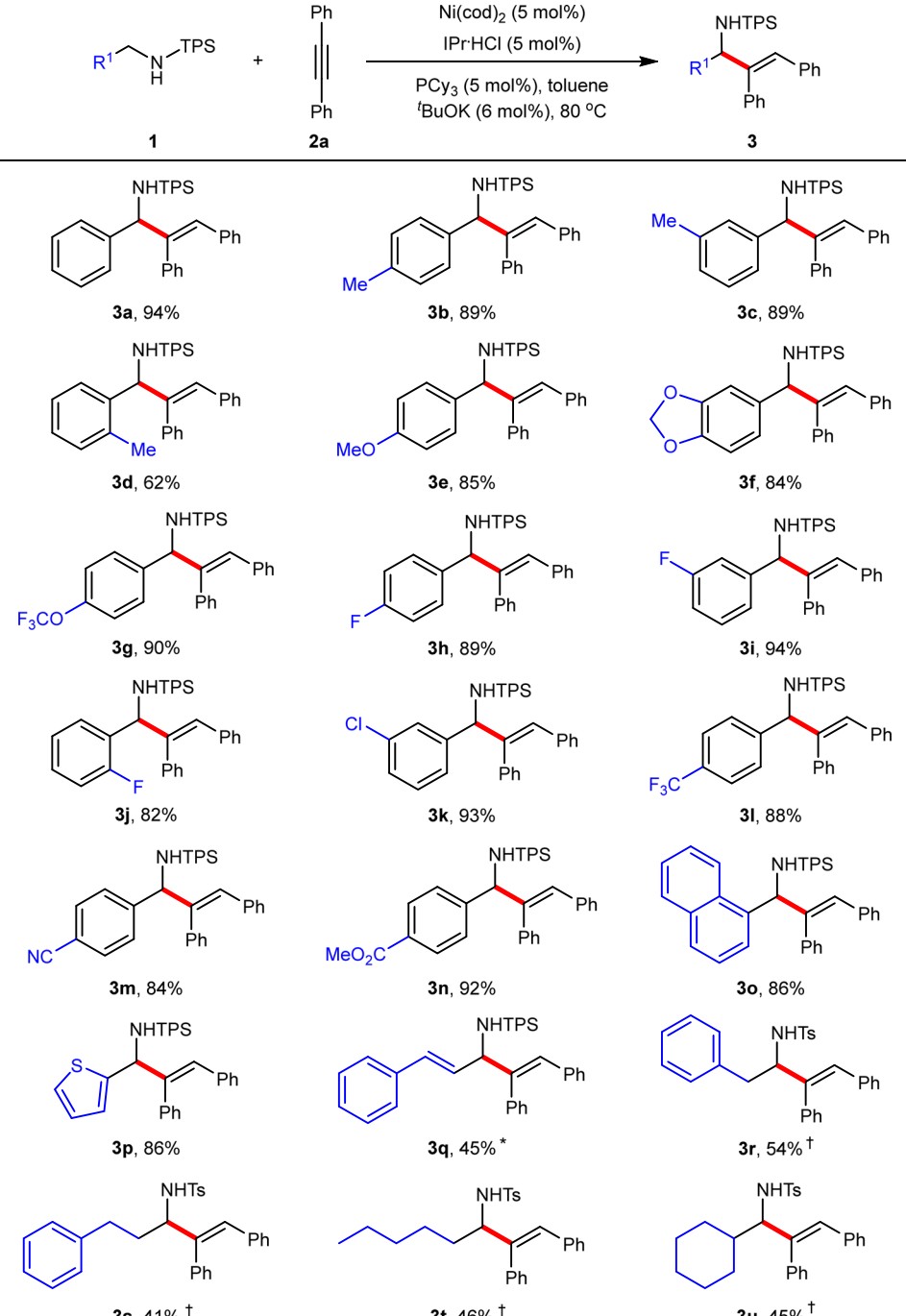

**Fig. 3 Scope of amines.** Reaction conditions: **1** (0.20 mmol), **2a** (0.22 mmol), toluene (2.0 mL) under $N_2$ for 1–12 h; yield of isolated products. *Ni(cod)$_2$ (10 mol%), IPr·HCl (10 mol%), PCy$_3$ (10 mol%), $^t$BuOK (12 mol%) at 110 °C. †Ni(cod)$_2$ (20 mol%), IMes·HCl (20 mol%), PCy$_3$ (20 mol%), $^t$BuOK (22 mol %) at 130 °C. Ts = tolylsulfonyl.

were well compatible with the current reaction, providing the corresponding products in 79−92% yields. Notably, 2-tolylalkyne gave a 1:1 mixture of *E*:*Z* isomers (**4c**), probably because the significant steric hindrance on the aryl ring forced the alkene to isomerize.

In contrast, electron-deficient groups such as OCF$_3$ (**4g**), F (**4h**), and CF$_3$ (**4i**) on the phenyl ring led to slightly lower yields even at a higher temperature. In addition, both dialkyl alkynes (**4j** and **4k**) and alkyl aryl alkynes (**4l**−**4p**) were well tolerated, providing both good yields and good to excellent regioselectivities. For example, 1-phenylpropyne gave 8.1:1 regioisomeric ratio (**4l**), and the change of methyl to ethyl significantly increased the

ratio to 20:1 (**4m**). Bulkier alkyls (**4n**−**4p**) or silyl (**4q**) led to a single regioisomer. However, non-symmetrical dialkyl alkyne (**4r**) cannot afford good regioselectivity probably owing to low differentiation between isopropyl and methyl groups.

**Reaction utility and mechanistic investigation.** To demonstrate the utility of the reaction, a gram-scale reaction of the model substrates was conducted under the standard conditions, affording the desired product **3a** in 88% yield, without significant loss of the yield (Fig. 5a). In addition, the formed allylic amine **3a** can act as a versatile synthetic intermediate to participate into various

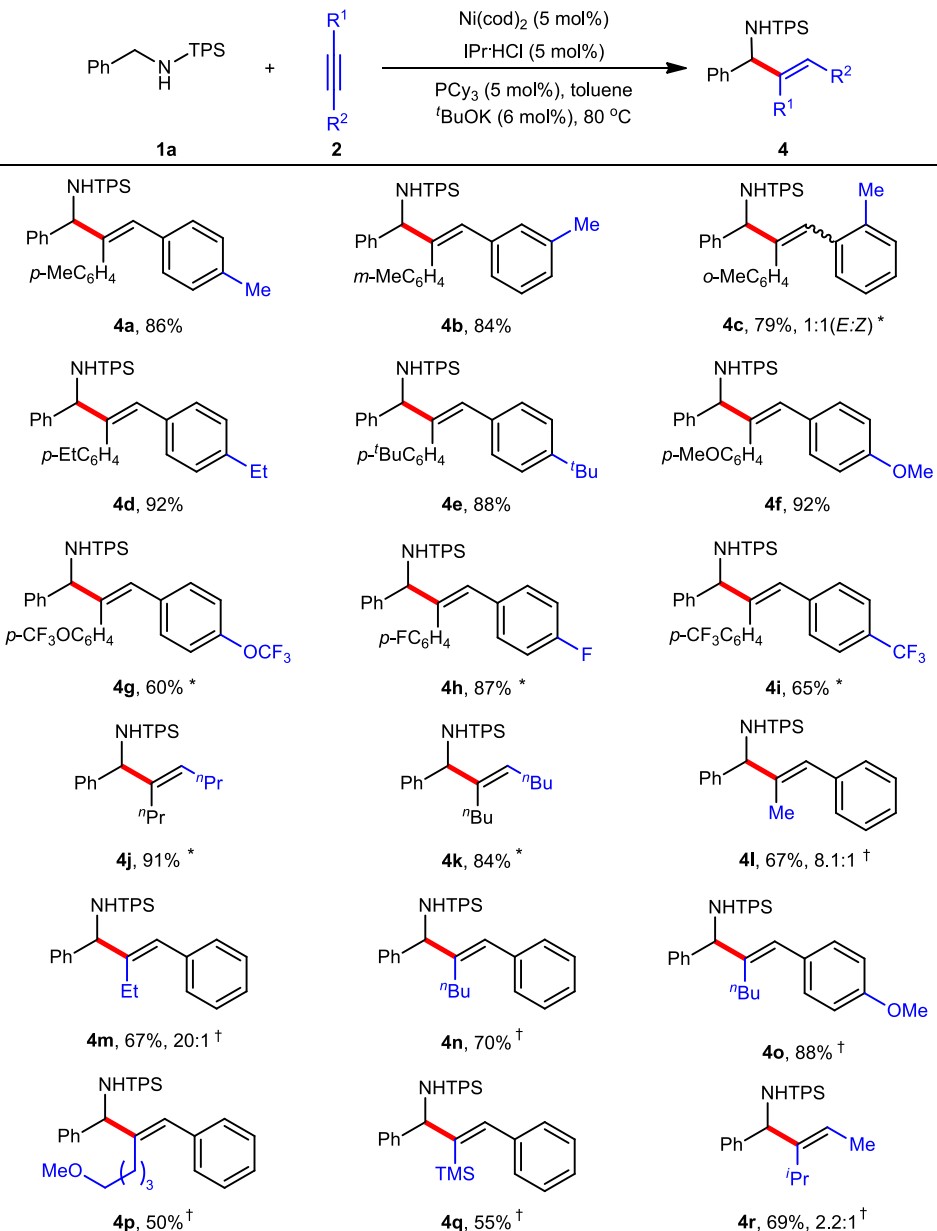

**Fig. 4 Scope of alkynes.** Reaction conditions: **1a** (0.20 mmol), **2** (0.22 mmol), toluene (2.0 mL) under $N_2$ for 1–12 h; yield of isolated products. *110 °C. †Ni (cod)$_2$ (10 mol%), IPr·HCl (10 mol%), PCy$_3$ (10 mol%), $^t$BuOK (12 mol%) at 110 °C and regioisomer ratio. TMS = trimethylsilyl.

transformations. For example, hydrogenation followed by typical deprotection protocol of the sulfonyl group provided compound **5** in 68% yield. Moreover, direct oxidation of **3a** resulted into a synthetically useful α-amino ketone **6** in 90% yield.

To gain insights into the reaction mechanism, some mechanistic experiments were carried out. Deuterium labeling experiments revealed that 100% allylic deuterium and 94% olefinic deuterium existed in product ***d*−3a**, together with deuterated *Z*-stilbene (12% yield). This result showed that there was partial deuterium-scrambling at the vinylic position, which may be ascribed to a reversible insertion of C=C double bonds in products or stilbene into N−H bonds (Fig. 5b). In addition, deuterated *Z*-stilbene was obtained, indicating that a part of alkynes were reduced during the reaction process. Crossover experiments between ***d*−1a** and **1e** suggested that the allylic and olefinic hydrogens may originate from different amide molecules (Fig. 5c), excluding an oxidative addition pathway. The observed kinetic isotopic effect ($k_H/k_D = 2.7$ in the intermolecular

competitive reaction and $k_H/k_D = 2.2$ in parallel reactions, Fig. 5d) implied that the cleavage of the benzylic C−H bond could be involved in the rate-determining step. Notably, in case of dimethylamino benzylic amide **1v** as the substrate, imine **1v′** was detected (Fig. 5e). Moreover, the competitive reaction between amide and the imine showed that both of them gave the corresponding products in comparable yields (see Supplementary Information).

These results suggested that an imine intermediate could be involved in the catalytic cycle. In addition, the stoichiometric reaction of a five-membered nickelacycle[42–44] and amide **1a** with or without IPr afforded the desired product **3b** in 68% and 9% yields, respectively, suggesting that both the nickelacycle and IPr were critical to the reaction (Fig. 5f). Based on these mechanistic experiments and previous literature reports[45–49], a possible reaction mechanism was proposed (Fig. 6). At the induction stage, the nickel-catalyzed transfer hydrogenation of alkyne **2a** with amine **1a** furnishes *Z*-stilbene and imine **1a′**. Then, **1a′**, **2a**,

**Fig. 5 Synthetic utility and mechanistic experiments. a** Gram-scale reaction and product transformation. **b** Deuterium labeling experiments. **c** Intermolecular competition. **d** Determination of kinetic isotope effect. **e** Detection of imine. **f** Stoichiometric reaction.

and the nickel catalyst undergo an oxidative cyclometallation to generate nickelacycle **B**, which is subsequently protonated by **1a**. The resulting intermediate **C** then proceeds through a direct intramolecular hydrogen transfer to give Ni−product complex **D**. Finally, catalyst transfer between **D** and **2a** occurs, releasing product **3a** and completing the catalytic cycle.

To further shed light on each individual elementary step of the catalytic reaction, we performed density-functional theory (DFT) calculations on the model reaction of *N*-benzylbenzenesulfona-mide and **2a** in the presence of a simplified Ni/NHC catalyst. At the induction stage (Fig. 7a), Ni−amine−alkyne complex **IN1** first undergoes a ligand-to-ligand hydrogen transfer (LLHT) via **TS1** with an activation Gibbs energy of 14.3 kcal/mol. The resulting intermediate **IN2** proceeds through a conformational change into its reactive form **IN3**. Then, another intramolecular hydrogen transfer occurs via **TS2** with an overall activation Gibbs energy of 18.1 kcal/mol, generating Ni−imine−alkene complex **IN4**. Finally, ligand exchange between **IN4** and **2a** takes place, leading to Ni−imine−alkyne complex **IN5** and *Z*-stilbene. At the product-formation stage (Fig. 7b), **IN5** first undergoes an oxidative cyclometallation via **TS3** with an activation Gibbs energy of 23.7 kcal/mol, generating nickelacycle **IN6**. Then, another amine substrate enters the catalytic cycle, forming hydrogen-bonded complex **IN7**. Subsequently, an intramolecular proton transfer occurs via **TS4**, leading to intermediate **IN8**, which then proceeds through a series of ligand exchange processes. After that, the resulting reactive isomer **IN9** transforms into Ni−product complex **IN10** via the turnover-limiting intramolecular hydrogen transfer via **TS5** with an overall

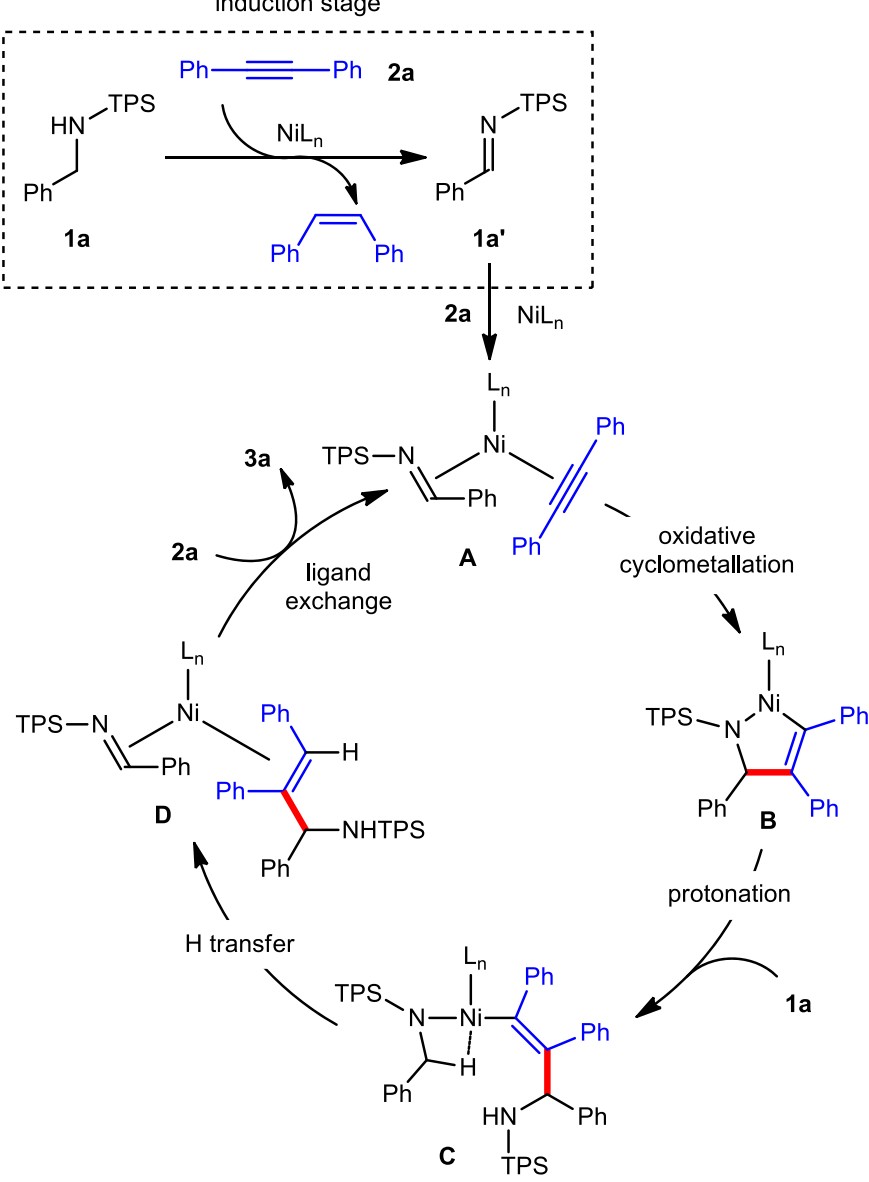

**Fig. 6 Proposed mechanism.** Induction stage and catalytic cycle.

activation Gibbs energy of 24.8 kcal/mol, which is in accordance with the observed kinetic isotopic effect (Fig. 5d). Finally, ligand exchange between **IN10** and **2a** takes place, releasing the final product and **IN5**, which triggers the next catalytic cycle. DFT calculations with TPS-protected substrate **1a** and IPr ligand indicated that the overall activation Gibbs energy is 22.4 kcal/mol. Replacement of TPS by Ts leads to a higher overall activation Gibbs energy of 24.9 kcal/mol. These results suggested that, as compared with the Ts group, a ca. 30-fold acceleration effect of the TPS group would be expected at 80 °C, which nicely reproduced the experimentally observed superior performance of the TPS protecting group (Fig. 2, entry 1 vs. entry 3). In addition, DFT calculations also suggested that the presence of PCy$_3$ may not reduce the overall activation Gibbs energy of the Ni-catalyzed reaction since replacement of NHC by PCy$_3$ did not promote the turnover-limiting hydrogen transfer step (see Supplementary Fig. 8). Instead, PCy$_3$ may act as an auxiliary ligand to facilitate the generation of the catalytic species and/or to inhibit catalyst deactivation.

## Methods

**General procedure for hydroaminoalkylation**. To a 15 mL pressure tube were added Ni(cod)$_2$ (2.75 mg, 0.01 mmol), IPr·HCl (4.25 mg, 0.01 mmol), PCy$_3$ (2.8 mg, 0.01 mmol), KO$^t$Bu (1.34 mg, 0.012 mmol), toluene (2.0 mL), alkynes (0.22 mmol) and amines (0.20 mmol) in a glove box. The tube was sealed with a Teflon cap and the mixture was stirred at 80 or 110 °C for 1–12 h. After cooled to room temperature, the crude product was filtered through a short pad of Celite, and the filtrate was concentrated under vacuum. The resulting residue was obtained by chromatography on silica gel column with petroleum ether/ethyl acetate as the eluent.

## Data availability

The authors declare that the data supporting the findings of this study are available within the article and its Supplementary Information file. For the experimental procedures and data of NMR, see Supplementary Methods in Supplementary Information file. For computed energies and Cartesian coordinates of the stationary points see Supplementary Data 1.

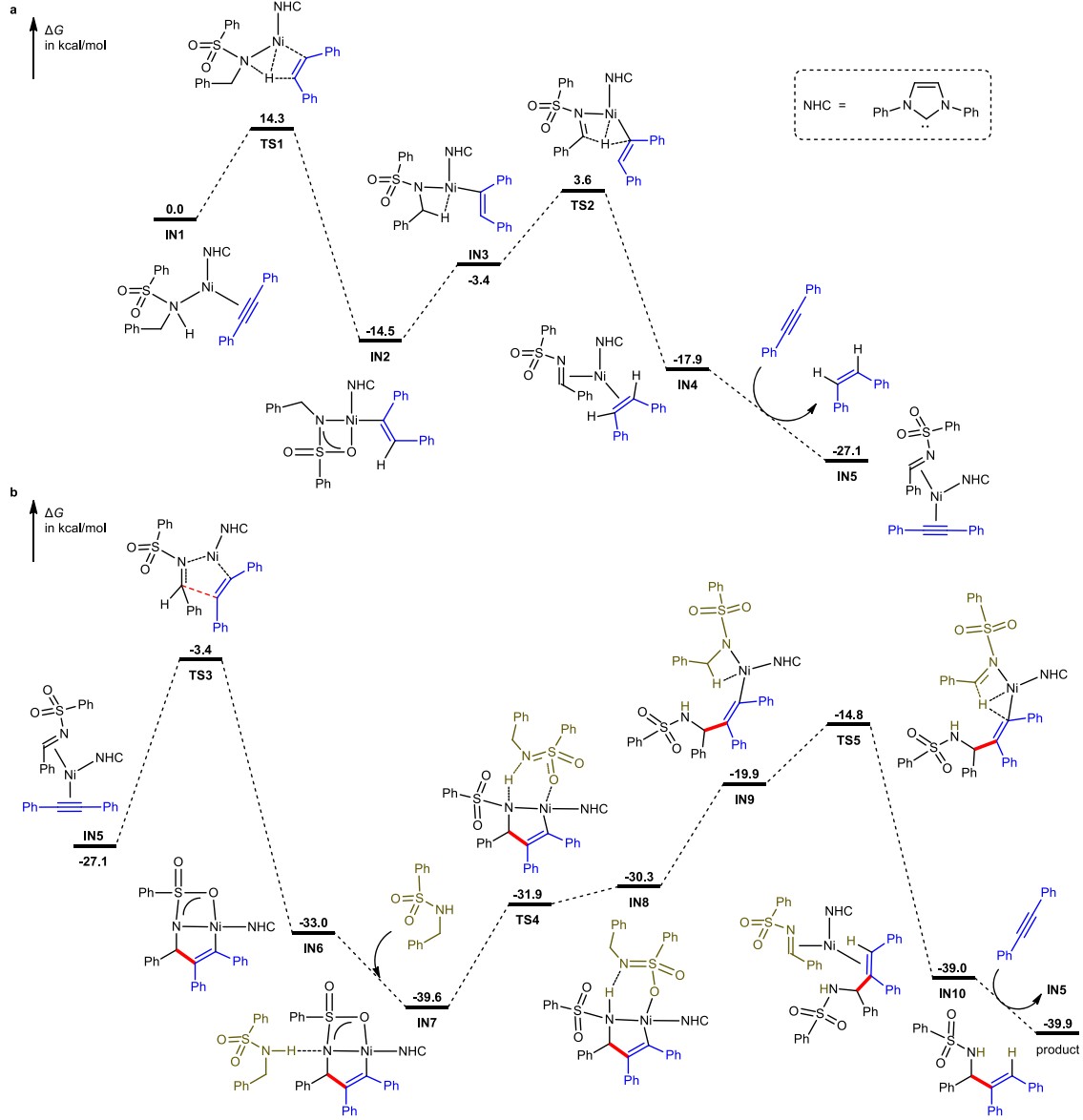

**Fig. 7 DFT calculations. a** Induction stage for imine formation. **b** Product-formation stage.

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

## Acknowledgements

This work was supported by the National Natural Science Foundation of China (21871145, 21933003, and 91856104), the Fundamental Research Funds for the Central Universities (63191601), High-Performance Computing Platform of Peking University, and National Supercomputing Center in Shenzhen (Shenzhen Cloud Computing Center).

## Author contributions

W.-W.Y. discovered and developed the reactions. R.L., H.C., M.-K.C. performed part of synthetic experiments. Y.-X.L., Y.W., Z.-X.Y. performed the DFT calculations and analyzed the computational results, Y.-X.L., M.Y. conceived, designed the investigations and wrote the manuscript. W.-W.Y. wrote the Supplementary Information.

## Competing interests

The authors declare no competing interests.

## Additional information

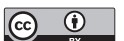 

