## [Peer Review File · Nature Communications]

REVIEWER COMMENTS

Reviewer #1 (Remarks to the Author):

This article from Ye and co-workers describes the intermolecular coupling of sulfonyl amines with alkynes to form allyl amines. A Ni(0) catalyst is used for this redox neutral coupling, with an NHC ligand being critical and a phosphine ligand serving as a beneficial additive. The transformation works with benzyl amines and, to a lesser extent, allyl and simple alkyl amines. The alkyne coupling partner can be dialkyl, diaryl, or unsymmetrical aryl/alkyl. The authors appropriately highlight a lot of precedent in the area of hydroaminoalkylation and related reactions, including some older reports on stoichiometric zirconium chemistry and more recent catalytic efforts. There are also several reports of related imine+alkyne coupling reactions that necessitate a stoichiometric reductant. The simple amine+alkyne coupling presented here-in is appealing compared to much of this work. Amines themselves are much more abundant and well behaved than imines, and the coupling is net redox neutral.

Some mechanistic experiments and DFT studies support an interesting mechanistic pathway. The amine itself being proposed to transfer a hydride to form the reactive imine in situ, which partially intercepts the more common imine+alkyne pathway for this reaction. In other words, the amine substrate serves as both the reducing agent and as an alkyne reservoir. The LLHT pathways proposed for both the induction step and for the proposed rate-limiting imine-forming step are both interesting and provide more support that this previously underappreciated elementary step can lead to interesting catalytic reactions. While the role of PCy₃ was not uncovered, its significant influence on the reaction may also stimulate further research and use of this strategy to facilitate Ni-catalyzed reactions. Overall, this article is a substantial contribution to an important research topic and warrants publication in Nature Communications. A few comments and suggestions are provided below.

1. The manuscript describes TS1 and TS2 but doesn't include these in the main manuscript, making the mechanistic discussion hard to follow with discussion split between Figure 6 and the SI.
2. The LLHT transition state 5 is interesting. Is there a reason the more intuitive beta hydride elimination/reductive elimination sequence is not proposed?
3. The intermolecular KIE study is confusing (fig 5D). In Figure 5C, a near identical experiment shows 100% benzylic D and 43% vinylic D, while 4D shows 27% D in both benzylic and vinylic positions. Is there a reason these two experiments give quite different results?
4. The manuscript needs to be checked more thoroughly for typos and errors. E.g. line 32 'alkyens.' Line 47 'ni.' Line 54 'protonic.' Line 137 'nickelacylce'

Reviewer #2 (Remarks to the Author):

Ye, Yu, and colleagues describe a nickel-catalyzed hydroaminoalkylation of alkynes with amines. This method provides rapid access to a series of synthetically useful sulfonyl protected allylic amines. The use of double ligand (IPr and PCy₃) for nickel catalytic system in combination with bulky sulfonyl group (TPS) protected amines are found as the key to promote the reaction effectively. The yields and functional group tolerance are good, the

substrate scope is considerably broad, and the regiocontrol levels are generally high. Additionally, they performed a detailed mechanistic study and DFT calculation to demonstrate the possible catalytic species, the rate-determining step, and the substitution effect of sulfonyl group (TPS) on the reaction outcomes. Although the current hydroaminoalkylation reaction is achieved in a racemic manner, it represents a rare and valuable C-H functionalization. I think this report is suitable for publication in Nature Communications after the following issues have been carefully addressed.

1. Potential byproducts derived from double bond isomerization, double bond walking, and dehydrogenation of allylic amine products would be generated in the presence of a nickel catalyst. Particularly for the product with alkyl groups on alkenes, double bond isomerization and double bond walking would readily occur and would be difficult to remove. Small peaks around 4.3 ppm in ¹H NMR appeared in several cases. I suggest the authors carefully check this and eliminate the possibility.
2. A deprotection of sulfonyl group to give allylic amine product would further enhance the synthetic utility for this attractive chemistry and is welcomed to include if possible.
3. The relatively lower yields were obtained in the cases of aliphatic amines. An explanation is encouraged.
4. The reactions shown in Scheme S9 and S10 are critical to the proposed catalytic circle. The authors should move these reactions to the manuscript from SI.
5. A typo: line 43, "alkyens" should be changed to "alkynes".

Point-to-Point Response

Response to Reviewer #1: Great thanks for the helpful comments

1. *“The manuscript describes TS1 and TS2 but doesn’t include these in the main manuscript, making the mechanistic discussion hard to follow with discussion split between Figure 6 and the SI.”*

Response: TS1 and TS2 have been added to Figure 6 in the revised manuscript.

2. *“The LLHT transition state 5 is interesting. Is there a reason the more intuitive beta hydride elimination/reductive elimination sequence is not proposed?”*

Response: In our DFT calculations on TS5, an alternative pathway via traditional β -H elimination followed by reductive elimination was also calculated. But even the Ni-H intermediate after β -H elimination was not located, indicating high energy of the expected stationary point and higher energy of the corresponding transition state for β -H elimination. In fact, although transition state for β -H elimination was not found, the un-converged structure already has higher electronic energy compared with TS5 ($\Delta E > 10$ kcal/mol). As a result, the LLHT pathway was proposed more favorable. And this explanation has been added into the supplementary information as Section 7.4.

3. *“The intermolecular KIE study is confusing (fig 5D). In Figure 5C, a near identical experiment shows 100% benzylic D and 43% vinylic D, while 4D shows 27% D in both benzylic and vinylic positions. Is there a reason these two experiments give quite different results?”*

Response: In both two experiments in Figure 5C and 5D, the ratio of benzylic deuterium can be used for comparison, while the ratio of vinylic deuterium is variable depending on reaction time and reaction conditions, because there is partial deuterium-scrambling at the vinylic position (see Figure 5B). We reasoned that a reversible insertion of C=C double bonds in products or stilbene into N-H bond may occur, leading to such deuterium-scrambling. To eliminate potential confusing, we revised the depiction on Figure 5B as *“Deuterium labeling experiments revealed that 100% allylic deuterium and 94% olefinic deuterium existed in product d-3a, together with deuterated Z-stilbene (12% yield). This result showed that there was partial deuterium-scrambling at the vinylic position, which may be ascribed to a reversible insertion of C=C double bonds in products or stilbene into N-H bonds (Fig 5b).”*

4. *“The manuscript needs to be checked more thoroughly for typos and errors. E.g. line 32 ‘alkyens.’ Line 47 ‘ni.’ Line 54 ‘protonic.’ Line 137 ‘nickelacylce”*

Response: We have thoroughly checked the manuscript and revised all errors.

Response to Reviewer #2: Great thanks for the helpful comments

1. *“Potential byproducts derived from double bond isomerization, double bond walking, and dehydrogenation of allylic amine products would be generated in the presence of a nickel catalyst. Particularly for the product with alkyl groups on alkenes, double bond isomerization and double bond walking would readily occur and would be difficult to remove. Small peaks around 4.3 ppm in ¹H NMR appeared in several cases. I suggest the authors carefully check this and eliminate the possibility.”*

Response: We have collected all these spectra and compared them together, and found that the impurity stays almost at the same position, showing a triplet peak at 4.31 (t, $J = 8.0$ Hz). In consideration of the fact there are no significant minor peaks in the aromatic region, instead, only some minor multiple peaks scattered in the region between 1.68 and 0.95, we reasoned that this impurity should not be isomers or derivatives of products, more likely, it is impurities contained in ethyl acetate solvent during column chromatography. We concentrated a large volume of ethyl acetate, and detected the obtained impurity by ¹H NMR. ¹H NMR spectrum showed the impurity was a complicated mixture with the characteristic peak at 4.31 ppm. Owing to trace amounts, purification and characterization of them was difficult, and we still cannot give an exact structure of the impurity.

2. *“A deprotection of sulfonyl group to give allylic amine product would further enhance the synthetic utility for this attractive chemistry and is welcomed to include if possible.”*

Response: In order to remove the protecting group to form allylic amines, we have tried many methods, but either material decomposition via deamination occurred under strong deprotection conditions, or low yield was given under relatively mild conditions. For example, the product **3a** was protected with Boc first, and then was subjected to Mg turnings in MeOH with ultrasound according to general literature procedure to generate TPS-deprotected Boc-amide in less than 20% yield. The difficult deprotection was ascribed to big steric hindrance of multiple-phenyl substituted product **3a**. While we found that when the rigid C=C bond was transformed into C–C bond by hydrogenation, the deprotection will become much easier according typical procedures, likely owing to bigger structural flexibility.

3. *“The relatively lower yields were obtained in the cases of aliphatic amines. An explanation is encouraged.”*

Response: The lower yield of aliphatic amines was ascribed to more difficult ligand-to-ligand H transfer in the rate-determining step (TS5), because of higher α -C–H bond strength of alkylamines than that of benzylamines and higher transition state energy. This explanation has been added to the manuscript, *“We reasoned that higher α -C–H bond strength of alkylamines than that of benzylamines and higher transition state energy may result in this situation.”*

4. *“The reactions shown in Scheme S9 and S10 are critical to the proposed catalytic circle. The authors should move these reactions to the manuscript from SI.”*

Response: Scheme S9 and S10 have been added into the manuscript as Figure 5f, and related comments have also been inserted into the text, “*In addition, a nickelacycle intermediate was synthesized and treated with stoichiometric amides in the absence or presence of IPr ligand, providing the desired product in 68% and 9% yield, respectively, which suggested a critical role of IPr ligand (Fig 5f).*”

5. “*A typo: line 43, "alkyens" should be changed to "alkynes".*”

Response: The error has been revised.

REVIEWER COMMENTS

Reviewer #1 (Remarks to the Author):

The authors have effectively addressed my comments and criticisms. I believe there is a small mistake introduced in Figure 6. The label 'TS5' is now written near the protonation step of the catalytic cycle, though I believe this is meant to represent the H transfer step as it did in the original submission.

Otherwise I consider the article to be suitable for publication.

Reviewer #2 (Remarks to the Author):

The authors have done sufficient jobs in revising this manuscript.

Point-to-Point Response

Response to Reviewer #1: Great thanks for the helpful suggestions.

1. *“The authors have effectively addressed my comments and criticisms. I believe there is a small mistake introduced in Figure 6. The label 'TS5' is now written near the protonation step of the catalytic cycle, though I believe this is meant to represent the H transfer step as it did in the original submission. Otherwise I consider the article to be suitable for publication.”*

Reponse: The error has been revised in Figure 6.